# Label Free, Lateral Flow Prostaglandin E2 Electrochemical Immunosensor for Urinary Tract Infection Diagnosis

**Antra Ganguly** [1] [iD], **Tahmineh Ebrahimzadeh** [2], **Philippe E. Zimmern** [3], **Nicole J. De Nisco** [2] [iD] and **Shalini Prasad** [1,*] [iD]

1    Department of Bioengineering, University of Texas at Dallas, Richardson, TX 75080, USA; Antra.Ganguly@utdallas.edu
2    Department of Biological Sciences, University of Texas at Dallas, Richardson, TX 75080, USA; tahmineh.ebraimzadehpirshahi@utdallas.edu (T.E.); Nicole.DeNisco@utdallas.edu (N.J.D.N.)
3    Department of Urology, University of Texas Southwestern Medical Center, Dallas, TX 75390, USA; Philippe.Zimmern@utsouthwestern.edu
*    Correspondence: Shalini.Prasad@utdallas.edu

**Abstract:** A label-free, rapid, and easy-to-use lateral flow electrochemical biosensor was developed for urinary tract infection (UTI) diagnosis in resource challenged areas. The sensor operates in non-faradaic mode and utilizes Electrochemical Impedance Spectroscopy for quantification of Prostaglandin E2, a diagnostic and prognostic urinary biomarker for UTI and recurrent UTI. To achieve high sensitivity in low microliter volumes of neat, unprocessed urine, nanoconfinement of assay biomolecules was achieved by developing a three-electrode planar gold microelectrode system on top of a lateral flow nanoporous membrane. The sensor is capable of giving readouts within 5 min and has a wide dynamic range of 100–4000 pg/mL for urinary PGE2. The sensor is capable of discriminating between low and high levels of PGE2 and hence is capable of threshold classification of urine samples as UTI positive and UTI negative. The sensor through its immunological response (directly related to host immune response) is superior to the commercially available point-of-care UTI dipsticks which are qualitative, have poor specificity for UTI, and have high false-positive rates. The developed sensor shows promise for rapid, easy and cost-effective UTI diagnosis for both clinical and home-based settings. More accurate point-of-care UTI diagnosis will improve patient outcomes and allow for timely and appropriate prescription of antibiotics which can subsequently increase treatment success rates and reduce costs.

**Keywords:** non-faradaic immunosensing; prostaglandin E2; electrochemical impedance spectroscopy; urinary tract infection; lateral flow electrochemical immunosensor; point-of-care UTI diagnosis; self-monitoring UTI dipstick

## 1. Introduction

Urinary tract infection (UTI) is defined as infection in any part of the urinary tract with a pathogen (mainly bacteria) resulting in inflammation [1,2]. UTIs are seldom fatal, but can cause high morbidity, and affect people of all demographics from neonatal to geriatric populations [1,3]. According to Center for Disease Control and Prevention (CDC) National Healthcare Safety Network (NHSN), UTI is among the most common health care related infections to be reported [2] and is one of the leading reasons for treatment in primary care settings [4]. UTI has a life-time incidence of 50–60% in adult women [5] and there is a 50–80% chance that a woman has UTI if she presents to a primary care clinic with typical symptoms [4,6]. About 7 million office visits, 1 million emergency room visits, and 100,000 hospitalizations related to UTI are reported annually [7]. The current gold standard for UTI diagnosis is lab-based culture of mid-stream clean-catch urine when clinical symptoms develop [4]. The diagnostic window of clinical urine culture is typically 2–3 days. This long wait time is primarily due to (i) the time required for

bacteria to grow in sufficient numbers for reliable detection and (ii) logistic delays in sample transportation from primary care clinics to centralized labs. Clinical urine culture results are influenced by the expertise of the personnel that can lead to diagnostic error rates of 30–50% [8]. Further, the diagnostic thresholds (i.e., species and colony forming unit per mL) are variable resulting in false negatives [4]. During the 2–3-day diagnostic window, either no therapy is initiated until the culture results are obtained, or empiric therapy is often initiated in which a broad spectrum of antibiotics is prescribed. Empiric therapy exposes the patient to unnecessary antibiotic costs and risks of allergy. Overuse and misuse of broad-spectrum antibiotics can lead to antimicrobial resistance (AMR) and treatment complication (if the choice of antibiotic ends up being inadequate for the growing bacterial strain) and should be avoided as the global AMR crisis deepens. This is especially important in the case for Asymptomatic Bacteriuria (ASB, i.e., bacteria in urine without symptoms) which usually requires no treatment [4,6].

UTI caused by multidrug resistant pathogens are among the most prevalent health challenges today [3,9]. Early detection and rapid diagnosis of UTI, independent of the causative pathogen is required to facilitate timely and appropriate administration of antibiotics. To this end, several high throughput automated diagnostic platforms such as Matrix Assisted Laser Desorption Ionization–Time of Flight Mass Spectrometry (MALDI-TOF MS), Reverse Transcription Polymerase Chain Reaction (RT-PCR) and urine flow cytometry have been developed in the recent years [3,10]. These techniques, though rapid, are not suitable for routine point-of-care (POC) testing at outpatient or homebased settings as they are expensive and require complex sample preparation, dedicated laboratory space, and trained professionals [10].

Adoption of the urine dipstick for assessing leukocyte esterase and nitrites, which suffers from poor specificity and a high false-positive rates, for UTI diagnosis, has inspired research for POC UTI diagnostics over the last four decades. To obviate imprecise empirical treatment, rapid and definitive POC biosensor technology has populated the research space in recent years [3]. There are a number of commercially available urinary dipstick biosensors which can be broadly classified as (i) culture-based devices, (ii) semi-automated analyzers, and (iii) enzymatic assays [10,11]. The culture-based devices suffer from turnaround times of 16–24 h and are therefore not, by definition, POC. The other two non-culture-based categories of UTI POC devices are sophisticated versions of the traditional qualitative urine dipstick and report the levels of typical parameters (viz., specific gravity, pH, leukocytes, nitrite, protein, glucose, ketone, urobilinogen, bilirubin, erythrocytes) [11] in a qualitative or semiquantitative manner.

In their review paper, Mach et al., delineate the attributes of a successful UTI POC biosensor. These include (i) the ability to definitively rule out infection; (ii) rapid POC timeframe to effect treatment planning; (iii) easy sample preparation with little to no intervention from the end-user; (iv) compatibility with urine matrix effect; (v) versatility across different pathogen profiles in different clinical scenarios [3]. In this regard, the new generation of biosensors based on micro- and nanotechnologies are gaining traction due their compact form factor and low power requirements, enabling POC screening. Lateral flow-based immunoassays (LFIA) and electrochemical immunosensors are particularly promising due to their low cost, which is especially important for resource challenged regions where UTI is prevalent and there is limited access to healthcare. However, LFIAs, in their traditional colorimetric formats, give qualitative results, lack sensitivity, and require sample preparation by the end-user making them unsuitable for routine, reliable monitoring.

To improve on the current state-of-the-art of UTI POC diagnostics, we propose a lateral flow based electrochemical biosensor dipstick that can diagnose UTI by quantifying the levels of Prostaglandin E2 (PGE2) in urine. Drawing parallels from the traditional electrochemical glucose biosensor for diabetes management, PGE2 was chosen as the single deterministic diagnostic and prognostic biomarker for UTI and recurrent UTI (rUTI) [9,12]. To avoid the complex sample preparation step, a single lateral flow membrane was used,

and the detection is done in very small samples (<100 μL) of neat (i.e., unprocessed and un-filtered) urine. The physiological range of urinary PGE2 is 500–4000 pg/mL [12]. Thus, for reliable detection, high sensitivity was achieved by using a powerful AC based electrochemical technique of Electrochemical Impedance Spectroscopy (EIS). Further, enhancement of sensitivity and improvement of binding kinetics was achieved through macromolecular crowding phenomena achieved through nanoconfinement of the biomolecules in the nanoporous lateral flow membrane [13–16]. To simplify measurement by the end-user, the use of labels such as redox tags or tracers was eliminated, and label-free sensing was achieved by operating in non-faradaic/capacitive mode. The proposed sensor gives read-outs in less than 5 min, enabling rapid UTI diagnosis for timely clinical decision making.

This work is the first demonstration of a lateral flow-based electrochemical biosensor for the quantification of urinary PGE2 in a label-free fashion for rapid, easy, and inexpensive UTI diagnosis. The current UTI diagnostics typically report the presence of bacteria and are often limited in the range of bacteria detected. The developed biosensor, being a non-culture based immunological sensor, is superior in this regard because it detects PGE2, which is produced in response to infection and is directly linked to the symptoms of UTI.

## 2. Materials and Methods

### 2.1. Materials and Reagents

Monoclonal α-PGE2 antibody was purchased from Arbor Assays (Ann Arbor, MI, USA). HPLC purified PGE2 was obtained in synthetic powder form from Sigma Aldrich (St. Louis, MO, USA). The crosslinker DSP (dithiobis (succinimidyl propionate)) and Phosphate Buffer Saline (PBS) were obtained from Thermofisher Scientific Inc. (Waltham, MA, USA). Artificial urine was prepared using the recipe (MP-AU or Multipurpose-Artificial urine) by Sarigul et al. and all the dilutions were prepared in deionized water [17]. The urine buffer pH was adjusted using sodium hydroxide (10% NaOH) and Hydrochloric acid (10% HCl) obtained from Sigma Aldrich (St. Louis, MO, USA). Pooled human urine samples (pH ~ 6.5) were obtained from Lee Biosolutions (St. Louis, MO, USA). The samples were aliquoted and stored at −20 °C until further use. For testing, the aliquots were thawed to room temperature and centrifuged. The supernatant was used for biosensing experiments. The lateral flow sensor system was constructed using Fusion 5 membrane from Cytiva (Global Life Sciences Solutions USA LLC, Marlborough, MA, USA).

### 2.2. Sensor Fabrication and SEM-EDAX Analysis

The sensors were constructed by depositing gold microelectrodes on Cytiva Fusion 5 lateral flow membrane by using a shadow mask in a cryo e-beam evaporator. The choice of the lateral flow membrane was done to ensure simple sensor fabrication to ensure low cost and easy and efficient sample handling. Fusion 5 was chosen as the single lateral flow membrane as it has fast wicking rates, requires no blocking, has low background noise, and does not require any additional lateral flow assay components (e.g., absorbent pad, blocking pad) for antigen detection [18–22]. The gold electrodes were patterned in the form standard planar three electrode electrochemical system (see Figure 1C). A thin layer of gold of ~1500 Å was achieved at a rate of 1 Å/s. EDAX analysis was done to characterize the fabricated sensor surface post deposition (see Figure 1D). Scanning electron microscopy experiments were done to study the porosity and pore density of the membrane (see Figure 1E,F). Gold deposition and SEM/EDAX analysis were done at UTD Cleanroom. The gold patterned lateral flow sensor was attached to a plastic adhesive backing for structural support (Adhesives Research, Glen Rock, PA, USA).

### 2.3. Electrode Modification and Sensor Stack Development

For the proposed PGE2 biosensor, highly specific monoclonal antibody has been used as capture probe. DSP has been used as the crosslinker for anchoring the monoclonal PGE2 antibody to the gold electrode surface. DSP is a water-insoluble, homo-bifunctional N-hydroxysuccimide ester (NHS ester) crosslinker that is thiol-cleavable, primary amine-

reactive, and has been widely used for immunosensor development [23–26]. The protocol has been adapted from a previous work by our group [25]. The sensor operates using the principle of affinity based biosensors. The PGE2 antigen expressed in a given unfiltered urine sample (depending upon the presence and severity of UTI) gets preferentially captured by the antibody. The protocol steps have been discussed in detail in Figure S6.

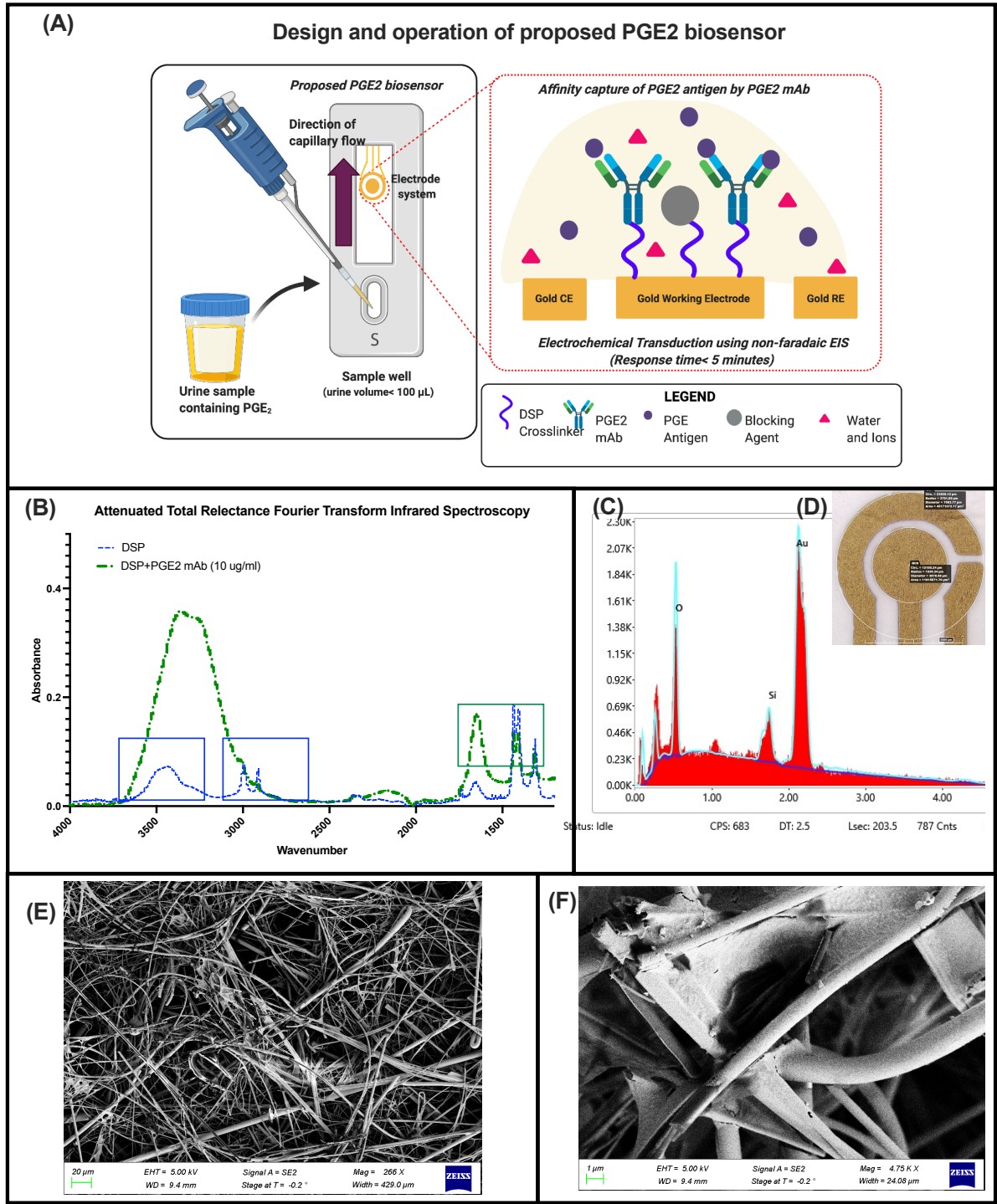

**Figure 1.** (**A**) Schematic showing the principle of sensor operation in non-faradaic mode and assay elements, (**B**) Attenuated Total Reflectance-Fourier Transform Infrared spectra of DSP crosslinker and DSP-monoclonal antibody conjugate, (**C**) EDAX characterization, (**D**) micrograph and (**E**,**F**) SEM images of gold deposited sensor on lateral flow membrane.

### 2.4. ATR-FTIR Studies

Figure 1A shows the schematic of the immunoassay stack of the non-faradaic biosensor. To validate the binding chemistry between the monoclonal PGE2 antibody (PGE2 mAb) and the thiol cross-linker (DSP), Attenuated Total Reflectance-Fourier Transform Infrared spectroscopy (ATR-FTIR) was carried out using Nicolet 6700 FTIR (Thermo Fisher Scientific, Waltham, MA, USA). Liquid samples of PGE2 mAb (10 µg/mL) conjugated to the DSP crosslinker (10 mM) analyzed for a wavelength range of 4000 cm$^{-1}$ to 600 cm$^{-1}$, at a resolution of 4 cm$^{-1}$ and 256 scans were collected. The obtained ATR-FTIR spectra have been shown in Figure 1B. Supplementary Table S1 lists the measured and expected peak positions characteristics of the molecules of interest.

### 2.5. Lateral Flow Optimization and Contact-Angle Studies

The flow characteristics of the base lateral flow membrane were evaluated using fluid wicking studies. Figure 2A,B show the wicking profile of the membrane in the down-web and cross-web directions. Based on these studies, the incubation time and the dispense volumes of the crosslinker, capture probe and the urine containing the target analyte were determined. The response time of the sensor was determined by these studies. To evaluate the hydrophilicity and surface wettability of the membrane, contact-angle studies were performed using a Ramé-Hart goniometer (Figure S2).

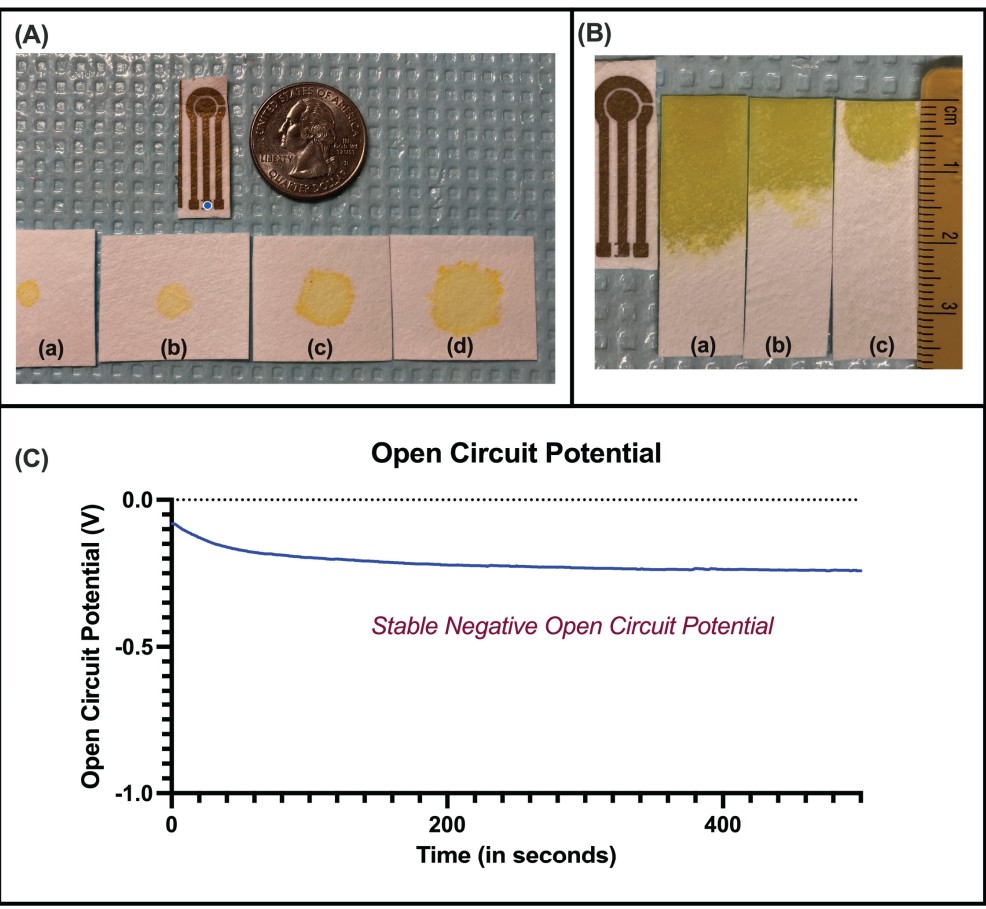

**Figure 2.** (**A**) Fluid wicking analysis to optimize fluid volume to cover working and counter electrode area for antibody immobilization and urine buffer dispensing. Fluid was dispensed on the substrate in increasing volumes (**a**) 0.5 µL, (**b**) 1 µL, (**c**) 2 µL, and (**d**) 5 µL. (**B**) Fluid wicking study in down-web direction to optimize sensor strip length (**a**) 60 µL, (**b**) 30 µL, and (**c**) 10 µL. (**C**) Open circuit potential measurement to establish stability of the developed sensor in urine buffer prior to PGE2 biosensing.

### 2.6. Open Circuit Potential Studies

The electrochemical stability of the three-electrode system was evaluated by measuring the Open Circuit Potential of the system in synthetic urine buffer medium (pH 6). In this study, a high-impedance voltmeter was connected in parallel to the working electrode (WE) and the reference electrode (RE) to measure the inherent potential gradient without applying any current (Gamry Instruments, Warminster, PA, USA). This was done to validate the electrochemical and thermodynamic stability of the lateral flow electrochemical biosensor in the presence of the highly ionic urine buffer prior to the calibration and performance testing of the PGE2 biosensor (see Figure 2C). OCP is widely used as preliminary test to identify the stability of the developed electrode when introduced to the highly complex body fluid matrices such as human urine. This was done to establish the time it takes for the bare gold electrode to achieve steady state after it has been introduced to the urine buffer microenvironment (time taken for the transient anodic and cathodic currents to reach equilibrium) [16,23,27].

### 2.7. Design of Electrochemical Impedance Spectroscopy (EIS) Experiments and Sensor Calibration in Human Urine

The EIS studies were performed using the Gamry Reference 6000 potentiostat (Gamry Instruments, PA, USA). Voltage of 10 mVrms was applied at the working electrode for a wide frequency range of 1 Hz–1 MHz. The impedance response for each change in concentration was recorded and extracted at 100 Hz (corresponding highest signal to noise ratio for non-faradaic capacitive behavior). The output is represented as the ratio change in impedance relative to the blank or un-spiked dose (zero dose). The PGE2 sensor was calibrated for 100–4000 pg/mL, which encompasses the physiological range of 500–4000 pg/mL in human urine. The EIS response of the developed sensor was studied in artificial/ synthetic urine and pooled human urine samples (Figures 3–5). Further the effect of urine buffer pH on sensor EIS response was studied (Figure 6A).

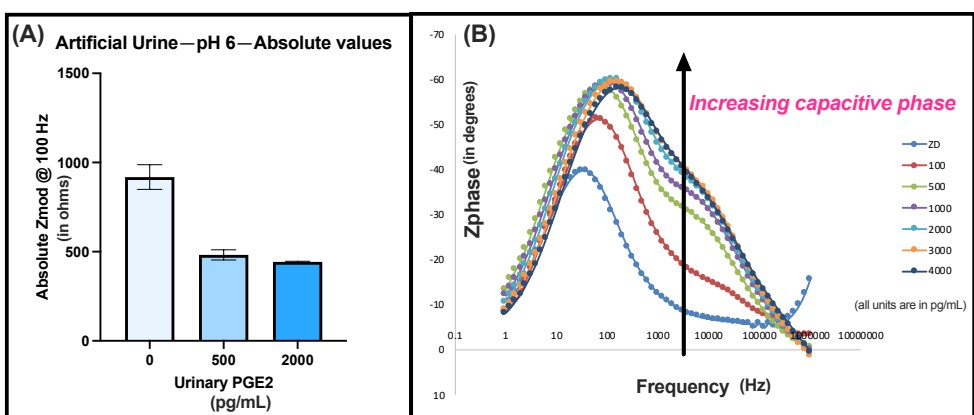

**Figure 3.** (**A**) EIS response of the developed sensor in synthetic/artificial urine (pH 6) and (**B**) Bode phase plot of the developed sensor showing a dose-dependent increase in capacitive phase in real pooled human urine samples.

### 2.8. Zeta Potential Studies

Dynamic light scattering experiments were conducted to measure the zeta potential of the biomolecules using Malvern Zetasizer NanoZS (Malvern Instruments, Malvern, UK). This was done to evaluate the variation of the surface charge of the antibody–antigen conjugate as a function of urine pH (low = 5, high = 7) and PGE2 dosing (low = 500 pg/mL and high = 5000 pg/mL). The pH adjustments were done using a pH meter (Accumet Research AR 20, Fisher Scientific). Zeta potential was calculated from the electrophoretic mobility by using the Smoluchowski equation [14,28–30].

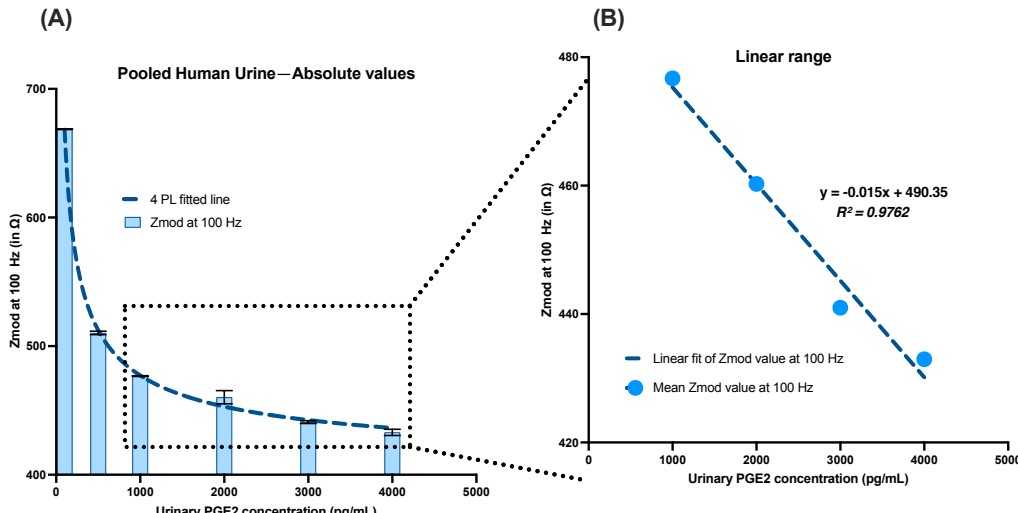

**Figure 4.** (**A**) Calibrated dose response curve for sensor response in pooled human urine using absolute Zmod values at 100 Hz (4 PL sigmoidal fitting) and (**B**) Calibrated dose response for linear operable range (linear fitting).

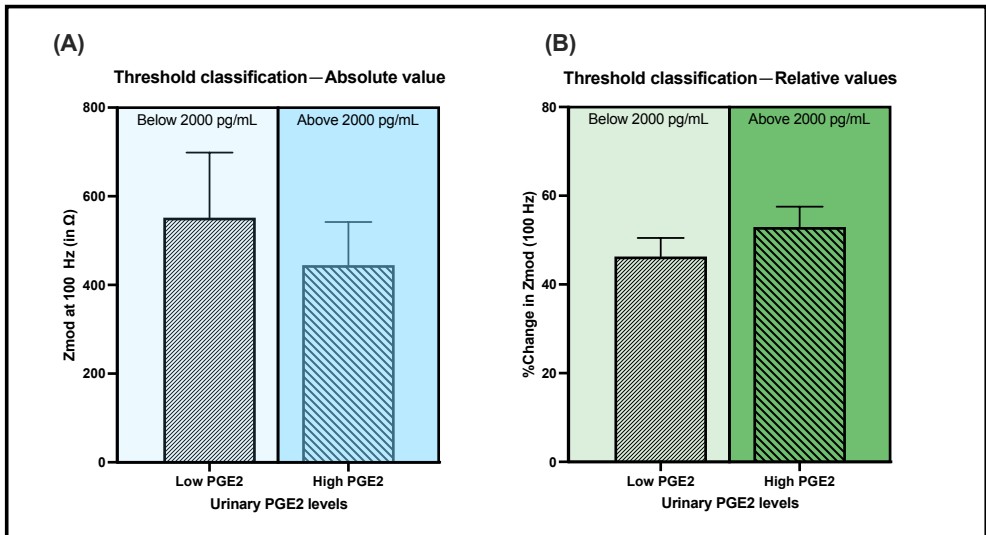

**Figure 5.** Threshold based UTI diagnosis using the developed sensor using EIS response for (**A**) absolute Zmod values at 100 Hz and (**B**) Zmod values relative to baseline dose.

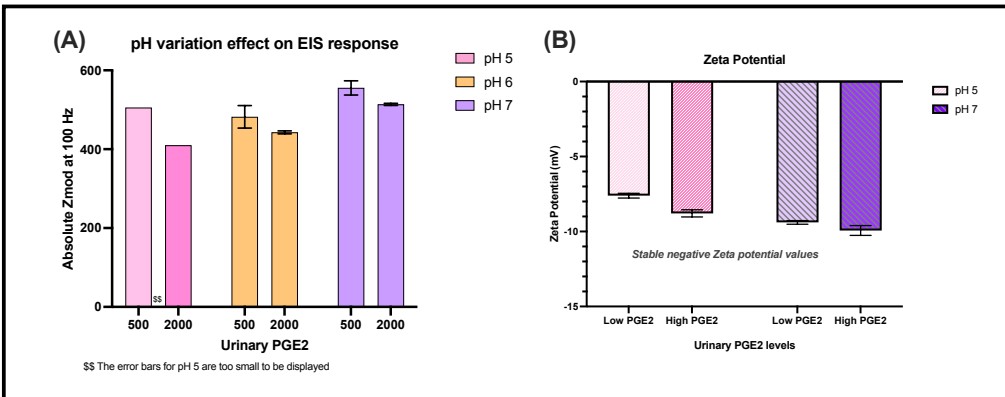

**Figure 6.** (**A**) Effect of pH on sensor response and (**B**) variation in zeta potential of the PGE2 antibody-antigen conjugate post binding at urine pH 5 and pH 7.

### 2.9. Sensor Performance in Varying Urine pH

The normal range of urine pH is 4.5–8.0. Urine is slightly acidic, with median pH of 6.0 to 7.5 [31]. Highly alkaline urine pH of 8.5 or 9.0 is indicative of urea-splitting organism and such a symptomatic patient with a high pH is directly diagnosed with UTI without further testing [31]. Keeping this in mind, the sensor was tested for urine pH values of 5, 6, and 7. The sensor was tested for zero dose (0 pg/mL), low (500 pg/mL), and high (4000 pg/mL) PGE2 concentrations for varying urine buffer pH.

### 2.10. Statistical Analysis

The error bars for calibrated dose responses for pooled and artificial human urine were calculated based on the root-mean squared error between the fitted sigmoidal 4 PL line and the mean value of signal response (absolute value or relative value of modulus of impedance, i.e., Zmod at 100 Hz) for $n = 3$ sensors (see Supplementary Information). For the threshold classification, the error bars were calculated based on the standard error of mean for the low (<2000 pg/mL) and high doses (>2000 pg/mL).

## 3. Results and Discussion

### 3.1. Immunoassay Analysis and Binding Chemistry Validation

The proposed PGE2 sensing scheme involves the affinity-based capture of PGE2 molecules expressed in urine using a highly specific monoclonal PGE2 antibody. The antibody is bound to the gold working electrode via strong thiol bonds using a DSP crosslinker. The formation of a self-assembled monolayer of the PGE2 monoclonal antibody (mAb) on the working electrode region is validated using ATR-FTIR analysis. When PGE2 is present in urine, it preferentially binds to the monoclonal antibody by affinity capture. The peaks specific to the DSP crosslinker and the binding between the DSP and PGE2 mAb have been identified.

The disulfide linkage in DSP chemisorbs rapidly to gold surface and forms strong and stable thiol bonds with gold electrode, while the active NHS groups on either end of DSP are reactive toward primary amine groups in PGE2 antibody. DSP contains an amine-reactive NHS ester at each end of an 8-carbon spacer arm which reacts with primary amine parts of the antibody (present in the side chain of lysine (K) residues and the N-terminus of each polypeptide) to form stable amide bonds eliminating N-hydroxy-succinimide [32].

Figure 1B shows the spectra for the DSP baseline and the DSP-mAb conjugate. Due to its primary amine structure, the antibody binds to the DSP crosslinker by cleaving the N-hydroxysuccinimide (NHS)–ester bond. This breakage of CO-NHS bond of DSP due to NHS binding of DSP linker to PGE2 mAb was validated by the decrease of the peak at 1735.05 cm$^{-1}$. Further, a shift in the primary amide peak from 1653.21 cm$^{-1}$ to 1646.98 cm$^{-1}$ as a result of binding and formation of stable secondary amide peaks at 1500–1550 cm$^{-1}$ was observed. In this way, the binding chemistry between the crosslinker and PGE2 mAb was validated. Supplementary Table S1 lists the peaks of DSP crosslinker and DSP-mAb conjugate optimized for biosensing.

### 3.2. Lateral Flow Optimization

Figure 1B,C highlights the crosslinked porous framework using scanning electron microscopy. This was done to measure the porosity and pore size of the lateral flow membrane (Cytiva Fusion 5 membrane), on which the gold electrodes are deposited [18]. The deposited electrodes conform to the texture of the underlying nanoporous membrane, which gives them a nanoporous surface and high surface roughness suitable for nanoconfinement of the assay biomolecules. This in turn increases the surface area of interaction between the capture probe and target urinary PGE2 molecules at the electrical double layer at the electrode–urine buffer interface. Due to macromolecular crowding, the interaction and binding kinetics between the molecules is increased which adds to the stability and sensitivity of the capacitive system [13–15,31]. This is critical for such small sample volumes (<100 μL) and low analyte concentrations (pg/mL concentration).

EDAX spectrum was collected to study the elemental composition of the lateral flow membrane and to confirm the successful deposition of the gold on the porous membrane. This is illustrated in Figure 1D. The broad peak at ~2 KeV corresponds to the successful gold deposition over the membrane forming the working, counter and reference electrodes. The other peaks correspond to the silicon (glass fibers), oxygen, carbon, nitrogen present in the lateral flow membrane.

The wicking profile of the membrane was studied in the horizontal (down-web and cross-web) and vertical directions as shown in Figure 2A,B. This was done to determine the volume of crosslinker, antibody, and the analyte required to cover the electrodes. The dimensions of the gold electrodes have been showed in the sensor micrograph in Figure 1C. The crosslinker and PGE2 mAb were immobilized on the working electrode where the specific binding interactions take place while the urine sample containing PGE2 was dispensed on both working and counter electrodes to complete the electrical circuit. From Figure 2A, a volume of 5 µL (c) was sufficient to cover the working electrode completely with the DSP crosslinker and the PGE2 mAb. From Figure 2B, it is evident that 60 µL of fluid covers the strip sufficiently well within the sensor response time of 5 min. In this way, these experiments helped optimize the wicking time of urine buffer and the dimensions (thickness 370 µm) [19] of the membrane for subsequent biosensing.

### 3.3. Electrochemical Optimization

#### 3.3.1. Electrochemical Stability Analysis

Prior to running the electroanalytical measurements, the thermodynamic stability of the sensor system was studied using Open Circuit Potential (OCP) measurements. It was observed that dynamic equilibrium was reached within 5 min (in less than 200 s) after the electrode system is introduced to the urine buffer medium. A steady state was reached, and the sensor response stabilized at a negative OCP magnitude of a few millivolts, indicating thermodynamic stability for subsequent electrochemical biosensing [23,32].

#### 3.3.2. Signal Enhancement Due to Nanoconfinement

The impact of signal enhancement was studied by comparing the impedance response of the developed nanoporous lateral flow electrochemical biosensor with a commercially available non-porous three electrode system (Metrohm, Riverview, FL, USA) for the same electrode dimensions and urine volume and the same assay protocol. Supplementary information Figure S1 shows the change in signal response from blank to low dose and compares the output response range. For the nanoporous membrane, a mean signal change of 45% was observed from the 0 to 100 pg/mL PGE2 dose whereas 2.5% was observed for 0 to 500 pg/mL for that of the commercial electrode (averaged over 3 intra-assay and 3 inter assay replicates). Further, the output response range was increased by 2–3 times for the proposed nanoporous system versus the non-porous (commercial) electrode system (see Figure S1).

#### 3.3.3. Surface Charge Behavior of Assay Stack

Zeta potential gives an indirect measure of a charge of the assay biomolecules suspended in urine buffer. Figure 6B shows the zeta potential of the PGE2 antibody–antigen conjugate post binding for pH 5 and pH 7 (low and high bookends for normal urine pH). The measured zeta potential values fall in the negative range (~$-7$ mV for the lowest pH) which indicates the stability of the conjugate (no agglomeration of biomolecules). As expected with increasing pH (pH 5 to pH 7), the zeta potential becomes increasingly negative. Next, for a given pH value, the zeta potential corresponding to the low to high dose values are not drastically different, since PGE2 is a minimally charged molecule. This further validates the fact that the signal response obtained in EIS experiments corresponds to the capacitive change resulting from change in dielectric constant of the EDL due to Ab–Ag binding and not due to the electrostatic noise from the urine buffer for a given pH.

### 3.3.4. Electrochemical Characterization of Sensor Response in Urine Buffer

Figure 3B shows the Bode phase plot in which the phase angle of the sensor impedance response is plotted as a function of logarithm of frequency. From the figure, it is evident that the phase response of the sensor becomes increasingly capacitive from low to high doses of PGE2 in urine. Further, maximum capacitive response (indicated by highest negative phase) is attained at nearly 100 Hz. Thus, 100 Hz was chosen to study the modulation of impedance for analysis and sensor calibration.

Figure S3 shows the Nyquist plot of the EIS data for sensor response in pooled human urine for PGE2 dosing. The Nyquist plot represents the imaginary part of impedance plotted on the y-axis versus the real part of the impedance on the x-axis. A typical semi-circle Nyquist plot is characteristic of non-faradaic mode of operation. In non-faradaic mode, since there is no redox labelling, the charge transfer resistance tends towards infinity which results in an incomplete semi-circle in the Nyquist plot [23,33]. In non-faradaic EIS systems, the imaginary part of the impedance is inversely proportional to the double layer capacitance. Thus, as the PGE2 level in the urine sample increases, the imaginary impedance reduces due to change in the dielectric constant at the double layer interface as a result of binding (see Figures S4 and S5). Thus, a dose-dependent change in the diameter of the semi-circle is observed as a function of dosing which validates that the dielectric properties at the electrical double layer interface are modulated due to the affinity capture of the PGE2 antigen in the urine by the highly specific monoclonal PGE2 antibody. Further as the surface concentration of PGE2 molecules increases, it results in decrease in the real impedance value due to the accumulation of biomolecules within the double layer which causes changes in conductivity of solution and rearrangement of water molecules and ions [23,33].

### 3.4. Performance Evaluation of Lateral Flow Electrochemical Sensor

The proposed sensor is constructed as a combination of lateral flow and electro-chemical immunoassays. Modulation of dielectric properties at the electrode–urine buffer inter-face and a rearrangement of ions and water molecules, due to Ab–Ag binding were captured using a powerful AC based electroanalytical technique of non-faradaic Electro-chemical Impedance Spectroscopy (EIS). In this technique, a small sinusoidal AC voltage is applied at the working electrode and a phase shifted current is obtained as the output. In EIS, the ratio of the input voltage to the output current gives a complex impedance characteristic of the biosensing system (correlated to the binding activity at a given analyte concentration) [34–37]. In EIS, the modulus of the impedance is reported at a particular frequency which was optimized at 100 Hz as described in the previous section. Through EIS studies, information regarding the capacitive makeup of the interface is analyzed to calibrate the sensor to obtain dose response curves. Calibrated sensor dose responses for PGE2 detection in synthetic urine (pH 6) is depicted in Figure 3A. Figure 4 shows the dose response curve for PGE2 detection in pooled human urine (n = 3, 4 PL fitting) for a wide dynamic range of 100–4000 pg/mL. The modulus of impedance decreases in a dose dependent fashion as the system becomes increasingly capacitive due to binding. A highly sensitive linear dose dependent response with an $R^2$ value of 0.9762 was obtained in the linear operable region of the 4 PL sigmoidal curve as depicted in Figure 4B. This has been discussed in further detail in Figure S7.

As shown in Figure 5A,B, it is possible to distinguish between low and high levels of PGE2 in pooled human urine. Thus, the sensor is capable of discriminating between UTI positive (above 2000 pg/mL) and UTI negative cases (below 2000 pg/mL) with significant robustness and reliability. The proposed sensor, in its current form, is intended for monitoring, preliminary testing or follow up and is not a standalone diagnostic test. As a proof of concept, 2000 pg/mL has been used as an arbitrary threshold for UTI classification based on the recent studies from Ebrahimzadeh et al. [12]. For patients with PGE2 levels below 2000 pg/mL, the likelihood of having a UTI is lower than that for patients with PGE2 levels at or above 2000 pg/mL. In patient samples with PGE2 levels above 2000 pg/mL, the

sensor might become a useful tool for clinicians to predict recurrence. The CV% averaged over inter-sensor replicates for the UTI positive and UTI negative results were found to be 4.57% and 12.16% respectively will fall well within the acceptable range of clinical standard practice (<20%) as per the Clinical laboratory Standards Institute (CLSI) guidelines [38,39]. In this way, the lateral flow electrochemical PGE2 biosensor was calibrated for urine PGE2 quantification.

*3.5. Effect of Urine pH and Matrix Composition on Sensor Performance*

Figure 6A shows the results of the evaluation of the effect of urine pH on the performance of the developed PGE2 biosensor. The pH of urine was varied between 5–7 and the response of the sensor was studied EIS. It was found that the sensor was able to reliably discriminate between the blank, low and high PGE2 levels for pH 5, 6, and 7. Hence, the proposed sensor response is capable of UTI diagnosis by reliably reporting PGE2 levels for the physiologically relevant pH range of human urine.

## 4. Conclusions

This work is the first technological demonstration of a lateral flow based electrochemical biosensor for PGE2 quantification in urine for UTI management. This work uses EIS to evaluate the sensitivity of the platform for a wide dynamic range of 100–4000 pg/mL for small volumes (<100 μL) of urine within 5 min. The surface of the lateral flow membrane was evaluated for nano-porosity using SEM and EDAX analyses. The binding chemistry of the antibody–antigen chemistry was validated using ATR-FTIR experiments. The surface charge behavior of the Ab–Ag conjugate was evaluated using dynamic light scattering experiments measured using zeta potential experiments. The performance enhancement of the sensor due to nanoporous surface morphology and macromolecular crowding was studied in comparison with commercially available analogous non-porous electrodes. The sensor was calibrated using non-faradaic EIS for urine pH range of 5–7 for synthetic urine (as control medium) and pooled urine. We envision the proposed sensing system as a vital clinical UTI diagnostic tool that will enable rapid, easy, and cost-effective POC tool which will enable clinicians to facilitate targeted interventions for better antibiotic stewardship while improving patient outcomes and reducing hospital visits and costs.

**Supplementary Materials:** The following are available online at https://www.mdpi.com/article/10.3390/chemosensors9090271/s1, Figure S1: Performance comparison of nanoporous and non-porous electrode substrates, Figure S2: Results for Goniometer experiments, Figure S3: Nyquist plot of EIS response in pooled human urine samples, Figure S4: Bode magnitude plot of EIS response in pooled human urine samples, Figure S5: Imaginary part of impedance vs frequency plot of EIS response in pooled human urine samples, Figure S6: Protocol steps for electrode modification and sensor stack development, Figure S7: Calibration dose response 4 PL sigmoidal fitting and linear fitting, Figure S8: Calibration dose response 4PL sigmoidal fitting for relative impedance values, Table S1: Observed and expected peaks of FTIR spectra.

**Author Contributions:** Conceptualization, S.P., N.J.D.N., P.E.Z., and A.G.; methodology, S.P. and A.G.; software, A.G.; experimentation and validation, A.G.; formal analysis, S.P., A.G., N.J.D.N., T.E., and P.E.Z.; resources, S.P. and N.J.D.N.; writing—original draft preparation, A.G. and S.P.; writing—review and editing, N.J.D.N., P.E.Z., T.E., and S.P.; visualization, A.G. and S.P.; supervision, S.P., N.J.D.N., and P.E.Z.; funding acquisition, N.J.D.N. All authors have read and agreed to the published version of the manuscript.

**Funding:** This work was funded in part by the Welch Foundation grant AT-2030-20200401 (N. J. De Nisco).

**Institutional Review Board Statement:** Not applicable.

**Informed Consent Statement:** Not applicable.

**Data Availability Statement:** The data presented in this study are available on request from the corresponding author.

**Acknowledgments:** The authors would like to thank Vikram Dhamu for support with SEM and goniometer experiments.

**Conflicts of Interest:** Shalini Prasad has a significant interest in EnLiSense LLC (Allen, TX, USA), a company that may have commercial interest in the results of this research and technology. The potential individual conflict of interest has been re-viewed and managed by the University of Texas at Dallas, and it played no role in the study design; the collection, analysis, and interpretation of data; the writing of the article; or the decision to submit the article for publication. The funders had no role in the design of the study; in the collection, analyses, or interpretation of data; in the writing of the manuscript, or in the decision to publish the results.

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
