# Peer review of "Label Free, Lateral Flow Prostaglandin E2 Electrochemical Immunosensor for Urinary Tract Infection Diagnosis"

_chemosensors, doi:10.3390/chemosensors9090271_

Round 1
Reviewer 1 Report
The introduction of the manuscript was well structured and stated several valid points to support the desirability of the label free immunosensors which are amendable to POC settings. It is necessary to point out that the various steps followed for the modifications of the electrode, its optimized parameters and the protocol followed for the various tests performed using the sensor is not sufficiently described, thus affecting the overall comprehensiveness of the manuscript. Also, the apparent presence of 2 or more linear ranges in the calibration curve questions the reliability of the observed results. Most importantly, the manuscript fails to provide the real/prototype of the said lateral flow assay (which should show the presence of an absorption pad) and instead exhibited a paper based Au modified electrode. Hence should change the term “lateral flow” to “paper based”. The manuscript should be considered for publications after a major revision which involves but not limited to the introduction of a section dedicated to the detailed steps of electrode modifications (including the role of the DPS cross-linker and the bonds formed), protocols for performing the tests and a valid mechanism to explain the change in impedance with increase in analyte concentration.
Comments:
- Include the word “using” in the sentence.
The lateral flow sensor system 131 was constructed using Fusion 5 membrane from Cytiva (Global Life Sciences Solutions USA 132 LLC, Marlborough, MA USA.
- Zmod at 2000pg/mL decreases (doesn’t not follow the 4PL curve fit) explain why?
- The Au is deposited on the surface of the Fusion 5 paper, and since a lateral flow assay includes an absorption pad which performs the whisking effect thus, please elaborate how the analyte that is whisked into the paper through the absorption pad. Or else consider removing the term lateral flow and use “paper based” instead.
- Absolute and relative method is classified into two ranges, namely, below, and above 2000pg/mL. Please explain how the sensor will perform and how to make sense of the resulting output at exactly 2000pg/mL.
- Electrode modifications steps are not mentioned, in particular the activation of the Au electrode surface to initiate thiol bond with the crosslinker and the immobilization step of the antibody. Furthermore, the optimization steps are unclear.
- The use of DSP cross-linker follows the interaction of the active NHS to interact with the NH2 group of PGE2 mAb and form an amide bond, however the binding to the Au electrode is far superior for reactive groups such as -SH as also mentioned by the authors that the antibody is bound to the gold working electrode via strong thiol bonds using a DSP cross-linker. A precise explanation as to the role of the DSP cross linker and how it achieves the thiol bond, and the amide bond is very much needed to improve the overall readability of the manuscript.
- The overall sensing range of 100-5000pg/mL as seen from the calibration curve in nonlinear. On closer inspection it seems to depict 2 or more linear ranges. Which is also evident in the bode plot whereby the increase in Z is nonlinear w.r.t analyte concentration. The fitting curve used in the study seemingly rises questions to the overall accuracy of the sensor.
- A section describing the steps followed for tests in urine samples must be included. Which should also include the accuracy of the developed sensor.
Reviewer 2 Report
The authors have described an interesting device for express testing of urinary infections based on immunodetermination of prostaglandin E2 as a universal biomarker. The detection is based on the combination of flow lateral test and non-faradaic electrochemical impedance spectroscopy. The immunosensor showed possibility to distinguish high and low levels of biomarker and demonstrated obvious advantages over conventional POC devices described in introduction. The manuscript can be recommended for publication after minor revision. 1. The use of open circuit measurements is confusing. (A) Au reference electrode should be first tested to show independence of its potential in time. (B) Sensitivity of the method is much lower than that of ESI and it cannot be used as a reference method. (C) The stability of the potential says nothing about performance of biosensor, moreover, it can be described as thermodynamic stability - there are no evidences of real equilibrium in the system and the authors themselves say abut steady-state conditions in the text. 2. More urine samples should be tested to monitor possible interferences. Or, please check biogenic thiols that could affect Au surface and potential. Some drugs and food additives like ascorbic acid and acetaminophen could affect redox potential and charge of the immunosensor interface. Technical notes: Abstract: UTI and PGE acronyms should be explained here Introduction: Page 2, line 59 - there is no need in MDR acronym - it is used only once. Line 63 - give explanations of MALDI-TOF, RT-PCR acronyms Figure 1A - legend text should be increased, check the use of DSP - it is introduced much later (Section 2.3). ATR-FT acronym should be introduced here Page 3, line 79 “F times” - please explain Page 6. Figure 3 - dimensions of the variables should be added to the axe legends. Page 7, Figure 4 - Fig. 4C shows the same as Fig. 4a and 4d as 4b - just add error bars to figs. A and B and remove Figs. C and D Section 2.9 and through the text - it seems better to mark as subscript ‘mod’ in Zmod variable.
Reviewer 3 Report
The manuscript chemosensors-1355369 entitled “Label free, lateral flow Prostaglandin E2 electrochemical immunosensor for Urinary Tract Infection diagnosis” describes de development of a lateral flow electrochemical immunosensor for the determination of Prostaglandin E2 (PGE2) using ESI as detection technique. The development of the electrochemical sensor is well-described and characterized, despite lack of novelty about the immunosensor development. The paper is well written and the results are interesting but some aspects should be revised. Major revision is required. Some important aspects are:
- Bode magnitude plot- Zmod Frequency should be include for each PGE2 concentration measurements, the same used for obtaining figure 4 (a) and (c).
- Please include which is the mathematical relation between Zmod and PGE2 concentration and the bondage of the adjustment.
- Interferences studied would enhance a lot the work, taking into account that it is based on a label free immunosensors and other molecules and proteins appear in the urine when the patient suffers from some other diseases.
- Limit of detection, quantification and dynamic range should be compared with other similar electrochemical immunosensors or other methods for detecting PGE2. Furthermore error bars are huge, so the measurement of precision as coefficient variation CV % should be express and compare among positive and negative results.
- There is a lack of optimization studies regarding the capture antibody immobilization over the developed electrodes, regarding incubation time, Ab concentration, etc.
- A curious aspect, that is not explain in the manuscript, is the reason why as the amount of PGE2 retain over the immunosensor surface increase (as increase the PGE2 concentration in the measured solution) the semi-circle in the Nyquist Plot showed at figure S3 decrease, indicating a reduction of the current resistance.
Round 2
Reviewer 1 Report
The author had made changes in the entire manuscript which improved coherence. Section 2.4 in particular needs re-arrangement as to describe the electrode modification first and then mention the antibody antigen interaction. Further, the linear range of 1000-4000 pg/mL was used as the operational range, hence it is indeed necessary to mention this while the author states “a wide dynamic range of 100-5000 pg/mL”. Additionally, the title should include “paper-based” to give a clear idea of the entirety of the sensor’s design and mention the intended use of the sensor is for “monitoring, preliminary testing or follow up and is not a standalone diagnostic test”.
Comments:
- Line 171 correct the spelling of biosensor
- Line 174 put “as” the crosslinker
- Line 199 “body fluid”
- Check and improve the overall English and correct typological errors throughout the manuscript
- Line 263 use “eliminating” instead of “and releasing”
- Include the explanation for the slight variation from 4PL fit line at 2000 pg/mL in the supplementary information under S8.
- Include “paper based” in the title
- Consider re-arranging section “2.4 Electrode modification and sensor stack development”. For better coherence the following sequence is recommended.
- Deposition of Au on paper (method)
- Role of DPS (stating the specific bonds formed between Au and how the antigen is anchored to the Au surface)
- Antibody antigen capture
9. A the author stated, “The linear portion of the graph was chosen as the operable region for analysis”. Hence, an operational range of 1000-4000 pg/mL is used in the study. Thus, consider clarifying the statement “a wide dynamic range of 100-5000 pg/mL” by including the detail of “with an operational linear range of 1000-4000 pg/mL”.
10. To ensure accurate intentions of the developed sensor, include “The sensor response is intended for monitoring, preliminary testing or follow up and is not a standalone diagnostic test” in line 403.
Reviewer 3 Report
T he authors have satisfactorily answered all my suggestions and questions.
Author Response
We greatly appreciate the reviewer’s input for improving the quality of the manuscript.